bioinformatics/chemical biology

jacalin, COVID-19, receptor-binding domain, *in silico* binding, hydroxychloroquine, human angiotensin-converting enzyme 2

**Author for correspondence:**
Veerappan Anbazhagan
e-mail: anbazhagan@scbt.sastra.edu

This article has been edited by the Royal Society of Chemistry, including the commissioning, peer review process and editorial aspects up to the point of acceptance.

# Targeting the glycan of receptor binding domain with jacalin as a novel approach to develop a treatment against COVID-19

Senthilnathan Rajendaran, Arunchalam Jothi
and Veerappan Anbazhagan

School of Chemical and Biotechnology, SASTRA Deemed University, Thanjavur,
Tamil Nadu 613 413, India

*In silico* analysis revealed that a lectin, jacalin from jackfruit seeds, recognizes a glycosylated region of the receptor-binding domain (RBD) of SARS-CoV2. Jacalin binding induces conformational changes in RBD and significantly affects its interaction with human angiotensin-converting enzyme 2. The result may open up exploration of lectin-based strategies against COVID-19.

## 1. Introduction

Coronavirus (CoV) belongs to a family of enveloped viruses that causes diseases in birds and mammals. The CoV was known in animals since 1930 and in humans since 1960. In humans, CoV causes mild respiratory tract infections, but recently identified human strains of the CoV are lethal, which includes SARS (severe acute respiratory syndrome-CoV), MERS (Middle East respiratory syndrome-CoV) and COVID-19 (SARS-CoV2) [1,2]. Among these three strains, the World Health Organization (WHO) declared the outbreak of COVID-19 as 2019–2020 coronavirus pandemic. The initial outbreak of COVID-19 was identified in Wuhan, China and then transported worldwide through international route. It infected millions of people and thousands have died as of August 2020. Compared to any other viral infection, the rate of human transmission caused by SARS-CoV2 is very high.[1] Currently, no antiviral drugs or vaccine is known against COVID-19. Thus, quarantine of the infected people, the lockdown of the states and imposing travel restrictions are suggested as an option to minimize

---

[1]https://www.who.int/emergencies/diseases/novel-coronavirus-2019/technical-guidance

the spread, which eventually disturb the economy [3]. Hence, the global health-care system urges the researchers to discover a solution against COVID-19.

CoV is a long single-stranded positive RNA virus with size ranging from 27 to 34 kilobases. The genome encodes for structural proteins and a non-structural replicase polyprotein. CoV contains four structural proteins, they are spike (S), envelope (E), membrane (M) and nucleocapsid (N) proteins [4]. The spike protein is a homotrimer, which plays an important role in viral entry into the host cells. Each spike monomer has S1 subunit and S2 subunit. The receptor-binding domain (RBD) present in S1 subunit recognizes and binds to host human ACE2 (angiotensin-converting enzyme 2) and triggers conformational changes in S2 to facilitate the membrane fusion and cell entry [5,6]. Therefore, RBD becomes a likely target for the development of vaccines, inhibitors and neutralizing antibodies. Several lines of evidence show that surfaces of human pathogens, including virus and bacteria, are evolved to be extensively glycosylated for structural and functional roles [7]. Interestingly, only N343 of RBD has modified glycosylation with N-acetylglucosamine (NAG) [8]. Thus, targeting the glycan of RBD may provide a key lead to develop targeted therapies.

Lectins are glycan-binding proteins, present ubiquitously in all forms of life. Some lectins are known for antiviral, antibacterial and anti-cancer activity [9]. The specific glycan-binding properties of lectins were explored in targeted drug delivery [10]. Jackfruits are abundantly available in India, and the seeds are a rich resource of jacalin. Jacalin, a 66 kDa homotetrameric lectin isolated from the seeds of jackfruit, was reported to block the HIV-1 infection of CD4 + lymphoblastoid cells [11]. Recently, we found that jacalin recognizes the bacterial cell surface glycan and enhances the antimicrobial activity of nanoparticles [12]. Jacalin capped platinum nanoparticles exhibit excellent antimicrobial activity and promote adaptive immune response in a host to suppress repeated bacterial infection [13]. Hence, we hypothesize that jacalin can be considered to develop targeted therapies against COVID-19. In view of the importance, we initiated a first investigation on the interaction of jacalin with RBD through an *in silico* approach. In addition, we studied the docking of the proposed drug, hydroxychloroquine (HCQ) with jacalin. The result indicates that jacalin recognizes the NAG of RBD and also has the ability to bind to HCQ, suggesting the potential of jacalin to carry targeted inhibitors or neutralizing agents.

# 2. Materials and methods

The crystal structures used for the analysis were downloaded from the protein data bank (PDB). The recently solved structure of SARS-CoV2 RBD complex with ACE2 receptor (PDB ID: 6LZG) was used in our study [8]. PDB ID: 1M26 of jacalin was used in the analysis [14].

Protein–protein docking was performed using HADDOCK and ClusPro 2.0 online servers [15,16]. We selected two different protein–protein docking tools to check the consistency of the prediction. ClusPro 2.0 follows three computational steps: (i) rigid body docking using the fast Fourier transform (FFT) correlation approach, (ii) root mean square deviation (RMSD)-based clustering of the structures generated to find the largest cluster that will represent the likely models of the complex, and (iii) refinement of selected structures [15]. HADDOCK is an information-driven, flexible docking approach that follows three computational steps: (i) rigid body energy minimization, (ii) semi-flexible refinement in torsion angle space, and (iii) final refinement in explicit solvent refinement [16]. The HADDOCK tool is one of the highly used and cited docking tools in a single year [15]. Similarly, ClusPro is also a standard tool with over 200 citations [16]. Moreover, both the methods are fully automated and web based and were ranked as the best methods of critical assessment of the prediction of interactions [17]. The active site/interface residues of RBD (343, 342, 339, 338, 368) and jacalin (1, 47, 121, 122, 123, 125, 78, 80) was given as input. Similarly, for the binding of RBD and hACE2, the residues of RBD (31, 19, 24, 34, 41, 42, 353, 83, 30, 38, 35, 82) and hACE2 (484, 475, 487, 453, 500, 446, 449, 498, 496, 486, 505, 417, 493, 502, 489) are given as input. The web server 'VERIFY 3D' was used to select the best model from the top cluster. The interacting residues were identified using Discovery Studio Visualization software and were also conformed using RING web server [18].

The secondary structure information for models was calculated using the web server '2Struc', which uses DSSP (Dictionary of Secondary Structure of Proteins), a gold standard approach for secondary structure assignments [19].

The structure of hydroxychloroquine (HCQ) drug is obtained from the PubChem database (PubChem ID: 3652). Molecular docking of HCQ was performed using AutoDock Vina with help of PyRx tool [20]. The grid box dimension of 22.5 × 22.5 × 22.5 Å for x, y and z, respectively, was built around the target for blind docking. All other parameters were kept as default. In addition to AutoDock, Dock Thor, a free web

server was used to cross-check the protein–ligand interaction. LigPlot+ program was used to identify the interacting residues between jacalin and HCQ [21]. Pymol was used to visualize the interaction between a lectin, jacalin and RBD of SARS-CoV2.

# 3. Results and discussion

The crystal structure of the novel corona spike RBD-ACE2 receptor (PDB ID: 6LZG) with 2.5 Å resolution was exploited in this work [13]. It consists of a concave RBD bound to claw-like structure on the surface of hACE2 (electronic supplementary material, figure S1). The presence of N-O bridge between R439 in RBD and E329 in ACE2 makes COVID-19-hACE2 complex more energetically favourable than the SARS-CoV-hACE2 complexes [5]. It is noted from the structure that hACE2 receptor has three glycation sites for the monosaccharide NAG, notably at N90, N53 and N322. RBD has one glycation site at N343.

In order to establish the interaction between the glycation site of RDB or hACE2 with jacalin, we used two different protein–protein docking servers, ClusPro 2.0 and HADDOCK (High Ambiguity Driven bimolecular DOCKing). To validate the docking procedure, the docking between hACE2 and RBD was performed (separately for both methods) and the predicted top model compared with the PDB structure (as it is docked). Interestingly, the top model selected based on the top-scoring function matches with PDB structure, and the residues involved in the interactions are similar (electronic supplementary material, figure S2). This supports that ClusPro 2.0 and HADDOCK are suitable for studying protein–protein interaction. From the crystal structure of jacalin-T antigen complex (PDB: 1M26) [14], the coordinates of T antigens were removed, and the amino acids around 4 Å of the sugar-binding site were allowed to dock with the glycan site of RBD or hACE2. After completing the docking, the top model was selected based on the least weighted HADDOCK score. The score obtained for jacalin–RBD complex and jacalin–hACE2 complex was −82.6 and −78.7, respectively. The HADDOCK docking was further verified by ClusPro 2.0. The weighted score (balanced model) obtained from ClusPro 2.0 for the jacalin–RBD and jacalin–hACE2 complexes was −1308.8 and −1184.5, respectively. It is clear from HADDOCK and ClusPro 2.0 analysis that the jacalin has slightly higher preference for RBD than hACE2, which may be useful in selectively targeting the SARS-CoV2.

The top-ranked docked model of jacalin–RBD is shown in figure 1. The generated model of jacalin–RBD complex was processed by RING 2.0 web server and D S Visualizer tool for the interaction and PyMOL tool for visualization. Figure 2 shows the H-bond interaction in the interface of RBD and jacalin. The H-bond analysis suggests that jacalin–RBD interaction through nine H-bonds (table 1). Noteworthy, NAG601 located in N343 of RBD could form H-bond with jacalin through S119 and D125 with a distance of 3.3 Å (electronic supplementary material, figure S3). It has been identified that the RBD–jacalin complex was stabilized hydrophobically by amide-pi stacked and alkyl interaction (electronic supplementary material, table S1). Interestingly, one electrostatic (salt bridge) interaction was noted between E340 of RBD and K2 of jacalin with 5 Å length. The electrostatic interaction is in the acceptable range of medium strength [22]. Besides these interactions, the RBD–jacalin complex was stabilized by van der Waals interaction, and the list of amino acids involved in van der Waals interaction is given in electronic supplementary material, table S2.

Further, the RBD glycan recognizing the ability of jacalin was verified by *in silico* mutation study. The NAG linked N343 of RBD was mutated to G343 and docking studies was performed as described above. As a result of a mutation in RBD (N343G), the HADDOCK score decreased from −82.6 to −52.9, suggesting that the binding affinity of jacalin to N343G RBD was reduced. A detailed analysis is present in table 2. It is noted from table 2, the electrostatic and van der Waals interaction decreased considerably. Interestingly, the buried surface 1359.4 Å$^2$ decreases to 1263.6 Å$^2$, indicating that RBD (N343G) was poorly recognized by jacalin. Similar results were obtained from ClusPro 2.0, where the score for a balanced model decreased from −1308.8 to −1157.4 and the score for the electrostatic-favoured model was affected significantly (table 2). The results suggest that jacalin recognizes the RBD using NAG in N343, which was facilitated by H-bond and electrostatic interaction between the amino acids present around the sugar-binding site of jacalin (electronic supplementary material, figure S3).

Having learnt of the jacalin preference to RBD, we extend the analysis on the impact of the jacalin on RBD–hACE2 complex. It is noted from the crystal structure, the NAGs of hACE2 were not involved in RBD-hACEs interaction [13]. Thus, the NAGs of hACE2 were not considered for docking jacalin–RBD complex with hACE2. Figure 3 shows the top model obtained from the docking of jacalin–RBD complex with hACE2. The score obtained for RBD–hACE2 was −151.0, whereas the score for the binding of jacalin–RBD complex to hACE2 was −131.0. Table 3 shows the binding parameter associated with RBD–hACE2 interaction in

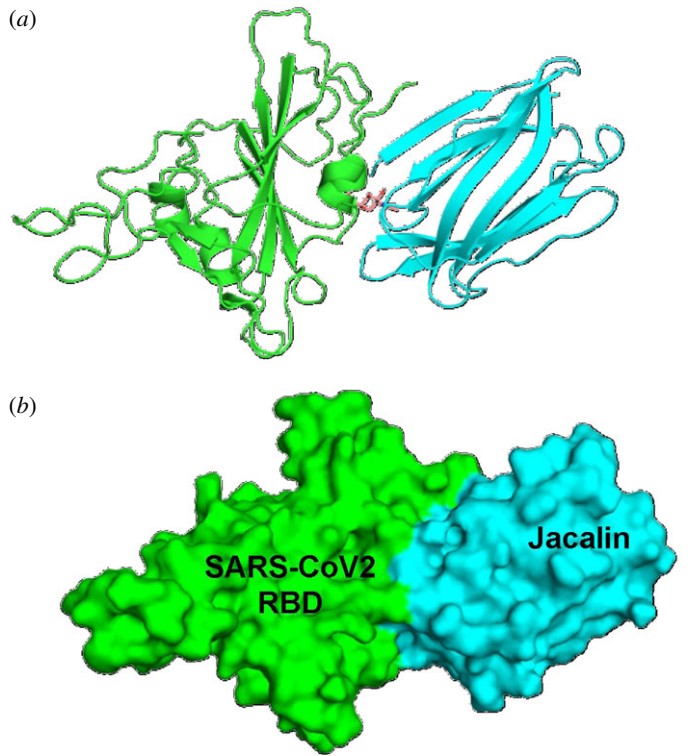

**Figure 1.** Top-ranked RBD–jacalin complex model from HADDOCK method. (*a*) Cartoon model, (*b*) surface model. RBD is shown in green colour and jacalin is shown in cyan colour. NAG linked with N343 on RBD is shown in stick model.

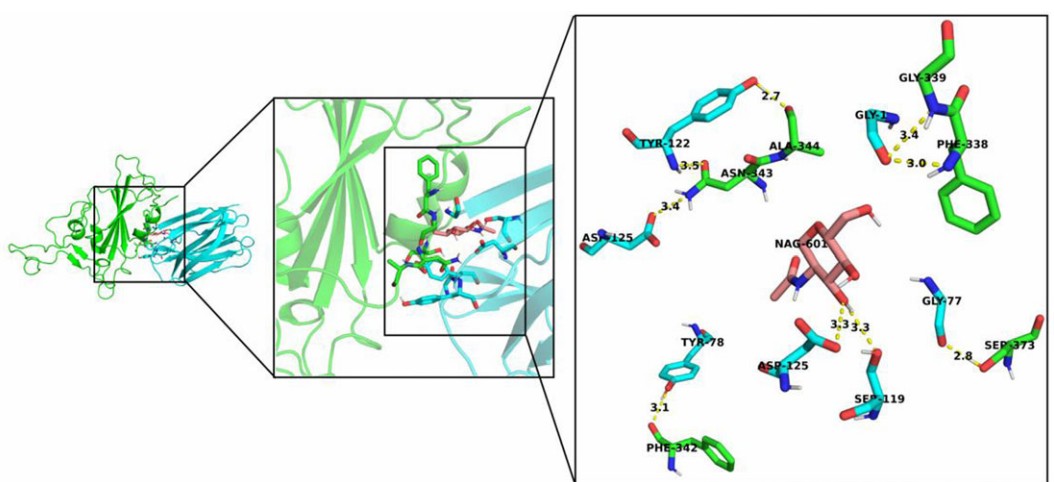

**Figure 2.** Visualization of hydrogen bonding interaction between RBD and jacalin complex.

the absence and presence of jacalin. It is noted from table 3 that jacalin affects the binding affinity between RBD and hACE2, as evidenced from the decreased buried surface area from 2026.1 to 1795.7 $\text{Å}^2$. Analysing the interaction of RBD–hACE2 and jacalin–RBD–hACE2 using DS visualizer suggests that H-bond and hydrophobic interaction between RBD–hACE2 is affected significantly in the presence of jacalin (electronic supplementary material, table S3 and figure S4).

The change in the secondary structure of the model was analysed by DSSP approach [19]. The α-helical, β-strand and coil content of hACE2 was 62.41, 3.52 and 34.06, and RBD was 15.38, 24.10 and 60.51, respectively, which is consistent with the crystal structure [13]. The docking of jacalin–RBD complex with hACE2 affects the secondary structure content of RBD: α-helical (11.28), β-strand (26.15) and coil (62.56), whereas the secondary structure of hACE2 was not affected significantly, α-helical

**Table 1.** Analysis of hydrogen bond interaction between RBD and jacalin. [D]—donor amino acid; C—chain; M—molecules involved; [A]—acceptor amino acid; d—distance.

| [D] | C | M | [A] | C | M | d (Å) |
|-----|---|---|-----|---|---|-------|
| Y78 | B | OH | F342 | A | O | 3.1 |
| Y122 | B | N | N343 | A | OD1 | 3.5 |
| N343 | A | ND2 | D125 | B | OD1 | 3.4 |
| Y122 | B | OH | A344 | A | O | 2.7 |
| S373 | A | OG | G77 | B | O | 2.8 |
| F338 | A | N | G1 | B | O | 3.0 |
| G339 | A | N | G1 | B | O | 3.4 |
| NAG601 | A | OH | D125 | B | OD1 | 3.3 |
| NAG601 | A | OH | S119 | B | OD1 | 3.3 |

**Table 2.** A detailed comparison of top docking model of RBD–jacalin complex and RBD (N343G)–jacalin complex by HADDOCK and ClusPro 2.0 method. RMSD—root mean square deviation from the overall lowest energy structure; $E_{vdw}$—van der Waals energy; $E_{elec}$—electrostatic energy; $E_{desol}$—desolvation energy; $E_{AIR}$—restraints violation energy.

| | RBD-jacalin complex | RBD (N343G)-jacalin complex |
|---|---|---|
| **HADDOCK 2.2** | | |
| cluster rank | 1 | 1 |
| cluster size | 322 | 399 |
| HADDOCK score[a] (arb. units) | −82.6 ± 2.2 | −52.9 ± 2.0 |
| RMSD | 1.3 ± 0.8 | 0.8 ± 0.5 |
| $E_{vdw}$ (kcal mol$^{-1}$) | −53.5 ± 6.1 | −29.2 ± 3.3 |
| $E_{elec}$ (kcal mol$^{-1}$) | −68.6 ± 17.0 | −55.6 ± 11.0 |
| $E_{desol}$ (kcal mol$^{-1}$) | −20.5 ± 5.1 | −19.3 ± 4.1 |
| $E_{AIR}$ (kcal mol$^{-1}$) | 51.0 ± 14.22 | 67.4 ± 2.60 |
| buried surface area (Å$^2$) | 1359.4 ± 103.4 | 1263.6 ± 75.8 |
| **ClusPro 2.0** | | |
| **models** | **weighted score (lowest energy)** | |
| balanced | −1308.8 | −1157.4 |
| electrostatic-favoured | −1320.9 | −1167.1 |
| hydrophobic-favoured | −1719.4 | −1723.0 |
| van der Waals + electrostatic | −208.8 | −193.2 |

[a]HADDOCK score = 1.0 $E_{vdw}$ + 0.2 $E_{elec}$ + 1.0 $E_{desol}$ + 0.1 $E_{AIR}$.

(60.90), β-strand (3.02) and coil (36.07). These results clearly indicate that the RBD conformational changes induced by jacalin-binding affects the interaction between RBD and hACE2.

In order to explore RBD recognizing property of jacalin in drug delivery, we analysed the interaction of jacalin with hydroxychloroquine. HCQ was considered as a model drug; moreover, HCQ was proposed for treatment against COVID-19 infection [23]. Previous studies suggest that jacalin binds to the variety of ligands beyond sugar, which includes porphyrin, ruthenium complexes, phthalocyanines, anti-cancer drugs, silver nanoparticles (NPs), platinum NPs (PtNPs), copper sulfide NPs (CuS NPs) and cadmium sulfide quantum dots [12,13,24,25]. Ability of jacalin to interact with HCQ was investigated by Autodock Vina server. The blind docking revealed that the jacalin binds to HCQ with a free energy of −5.8 kcal mol$^{-1}$, and the inhibition constant is 65.25 µM. The docking was further verified by DockThor, a free web server. The free energy value (−7.9 kcal mol$^{-1}$) obtained by DockThor is comparable to Autodock

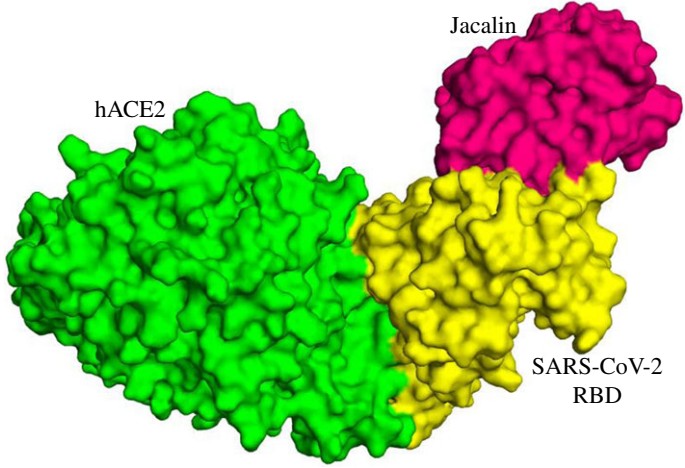

**Figure 3.** Surface model obtained for the docking of Jacalin–RBD complex with hACE2. Green—hACE2, yellow—spike protein RBD, pink—jacalin. NAGs of hACE2 are not involved in the protein–protein interaction.

**Table 3.** Binding parameters derived for the interaction of RBD–hACE2 in the absence and presence of jacalin.

|  | RBD–hACE2 | jacalin–RBD complex–hACE2 |
|---|---|---|
| cluster rank | 1 | 1 |
| cluster size | 400 | 354 |
| HADDOCK score[a] (arb. units) | $-151.0 \pm 0.4$ | $-131.0 \pm 1.5$ |
| RMSD | $0.4 \pm 0.3$ | $0.6 \pm 0.4$ |
| $E_{vdw}$ (kcal mol$^{-1}$) | $-68.2 \pm 2.2$ | $-59.7 \pm 4.2$ |
| $E_{elec}$ (kcal mol$^{-1}$) | $-283.3 \pm 8.6$ | $-217.8 \pm 38.6$ |
| $E_{desol}$ (kcal mol$^{-1}$) | $-26.7 \pm 0.8$ | $-27.9 \pm 9.4$ |
| $E_{AIR}$ (kcal mol$^{-1}$) | $5.5 \pm 0.64$ | $1.9 \pm 1.55$ |
| buried surface area (Å$^2$) | $2026.1 \pm 59.1$ | $1795.7 \pm 106.4$ |

[a]HADDOCK score = 1.0 $E_{vdw}$ + 0.2 $E_{elec}$ + 1.0 $E_{desol}$ + 0.1 $E_{AIR}$.

Vina server, suggesting the interaction between jacalin and HCQ. The free energy values suggest stronger binding, which is comparable to those observed for the interaction of carbohydrate and lectin [26]. Jacalin is known for binding to various galactose derivatives through hydrophobic interaction with a free energy range from $-2.9$ to $-7.8$ kcal mol$^{-1}$ [27]. Analysing the interaction between jacalin and HCQ from the top docked model indicate that F127, S128 and M129 form H-bond with HCQ. In addition, hydrophobic interaction was identified between HCQ and T72, V81, F104, L106, D125 and Y126. These interactions are presented as ligplot in figure 4 and suggest that jacalin has the binding pocket for hydrophobic drugs, which can be used for the development of jacalin-based drug carrier to target the RBD.

To support the global research effort of finding solutions for COVID-19 drives us to propose jacalin, because the recent study showed that jacalin-CuS NPs (JCuS NPs) eliminate the bacterial infection in zebra fish model [12]. The bacterial cell surface glycan recognition activity of jacalin plays an important role in targeting the antibacterial NPs. In another study, we found that jacalin-PtNPs (JPtNPs) not only rescue zebrafish from bacterial infection but also protect it from re-infection after many days. The results showed that JPtNPs protect the animal from bacterial infection through modulating pro-inflammatory cytokines and promote bacteria-specific antibody response to sustain repetitive infection [13]. Pioneering work by Corbeau *et al.* showed that jacalin binds to HIV-1 envelope glycoprotein (gp120) and its precursor (gp160) and inhibits the HIV-1 infection of CD4 + lymphoblastoid cells [11]. The spike protein of SARS-CoV2 is a glycoprotein, and jacalin recognizes RBD through glycosylation at N343 and has the ability to interact with drug. Based on the previous reports and present *in silico* study, we propose that the jacalin recognition of RBD may act as an inhibitor or be useful in developing targeted drug delivery vehicle against COVID-19.

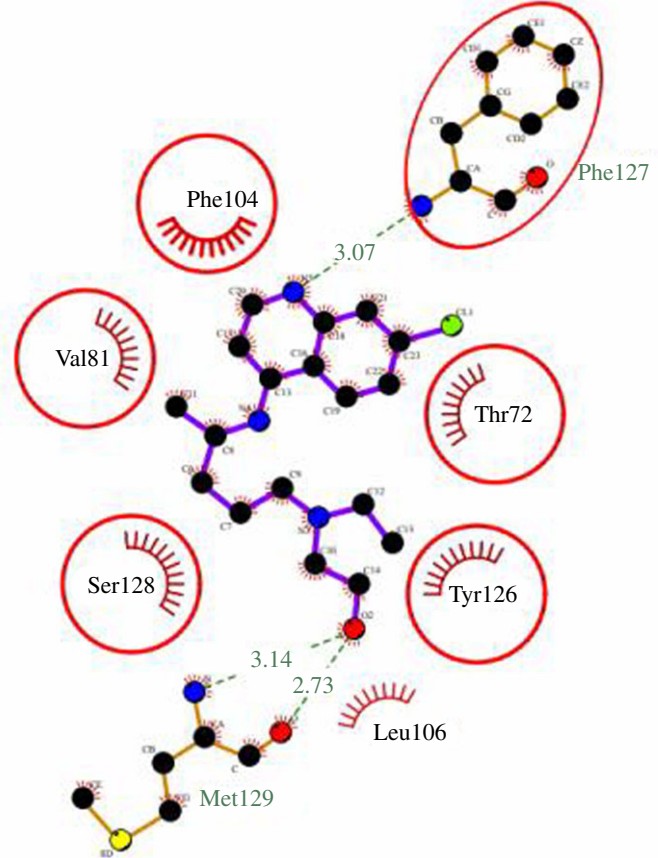

**Figure 4.** Ligplot shows the binding of HCQ with the jacalin.

## 4. Conclusion

The interaction between a lectin, jacalin and RBD of SARS-CoV2 was reported for the first time. A detailed *in silico* analysis suggests that jacalin effectively binds to RBD through NAG present in N343. As a result of jacalin binding, the interaction between RBD and hACE2 was affected significantly due to the change in the conformation of RBD. These results are promising because weakening the interaction between RBD and hACE2 may prevent the viral entry. Further, molecular dynamics and experimental investigation with diverse lectin against SARS-CoV2 may open up newer therapeutic options against viral infection.

Data accessibility.   Dryad: https://doi.org/10.5061/dryad.9kd51c5dk [28].

Authors' contributions. S.R. collected the data and analysed; A.J. analysed the data; V.A. conceptualized the work. The paper was written through the contribution of all the authors.

Competing interests. We declare we have no competing interests.

Funding. No specific funding to disclose.

Acknowledgements. S.R. thanks the Council of Scientific and Industrial Research, India for providing senior research fellowship (File No. 09/1095(0041)/19-EMR-I).

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
