## [Reviewer comments · Royal Society Open Science]

Review History

RSOS-200844.R0 (Original submission)

Review form: Reviewer 1

Is the manuscript scientifically sound in its present form?

No

Are the interpretations and conclusions justified by the results?

No

Is the language acceptable?

Yes

Do you have any ethical concerns with this paper?

No

Have you any concerns about statistical analyses in this paper?

No

Recommendation?

Major revision is needed (please make suggestions in comments)

Comments to the Author(s)

In the present manuscript, the authors have tried to verify whether the jacalin can be used as an inhibitor against RBD of Covid-19. Nevertheless, the manuscript lacks in the following aspects:

1. The two docking methods to be compared and whether there are significant changes in the predicted conformations between the top models to be discussed.
2. In the docked complexes, NAGs of hACE2 completely moved away from the protein. Is this the reason for the observed conformational changes rather the binding of jacalin? This might seriously affect the calculated binding affinities. Moreover, the authors need to justify the presence of NAG during docking between hACE2 and RBD-jacalin complex.
3. What are the validation methods followed to choose the right complex apart from docking scores? (both in the case of protein-protein and protein-ligand complexes).
4. Further the authors may dock hACE2 with RBD and compare with the PDB structure to validate the docking protocol.
5. From the final jacalin-RBD-hACE2 complex, it is understandable that jacalin does not interact with the interface between RBD-hACE2 complex. Then, how do the authors justify its efficiency to prevent the interaction of viral protein with host?
6. Are the authors sure that the interactions found would be stable? Molecular dynamics simulation might help the authors to assess the stability of the predicted complexes.

Review form: Reviewer 2

Is the manuscript scientifically sound in its present form?

No

Are the interpretations and conclusions justified by the results?

No

Is the language acceptable?

Yes

Do you have any ethical concerns with this paper?

No

Have you any concerns about statistical analyses in this paper?

No

Recommendation?

Major revision is needed (please make suggestions in comments)

Comments to the Author(s)

This study is quite interesting wherein the authors have tried to dock jacalin with RBD which is in direct contact with hACE2. However, Jacalin was also docked with Hydroxychloroquine. Here onward the study becomes quite laborious to understand

to begin with:

-> The abstract is too short, which does not give any details about the findings

->Regarding the graphical abstract it is mentioned as COVID19 RBD interacting with hACE2.

This is incorrect. Actually COVID19 is the infection and the protein is SARS-Cov-2. So the authors should have mentioned it as SARS-Cov-2 instead

->The statement like "The interaction between a lectin, jacalin and RBD of COVID-19 Coronaviruses (CoV) belongs to a family of enveloped viruses that causes diseases in birds and mammals" is incorrect

->With respect to Materials and methods, it is too short and does not explain the different parameters like active site/interface residues and their reference for protein-protein and protein ligand interaction. Even the grid box for Autodock software is not mentioned in the manuscript. The downloaded 3D structure has no structure validation report.

Regarding Result and discussion:

Authors have added literature review in this section also which need to be removed.

In order to establish the interaction between the glycation site of RDB or hACE2 with jacalin, we used two different protein-protein docking servers, ClusPro2.0 and HADDOCK (High Ambiguity Driven bimolecular DOCKing). ClusPro2.0 performs three steps computational, (i) rigid body docking using the Fast Fourier Transform (FFT) correlation approach; (ii) Root Mean Square Deviation (RMSD) based clustering of the structures generated to find the largest cluster that will represent the likely models of the complex; (iii) refinement of selected structures.

-> this statement is quite confusing.

->Within the Result and discussion session, software details are not required. This should be shifted to materials and methods section

-> Regarding the the score obtained for jacalin-RBD complex and jacalin-hACE2 complex was -82.6 and -78.7, respectively the authors haven't mentioned the +/- value of the HADDOCK score and the significance of the buried surface value

-> The weighted score (balanced model) obtained from ClusPro 2.0 for the jacalin-RBD and jacalin-hACE2 complexes were -1308.8 and -1184.5, respectively.

->Basically Cluspro only provides the docked poses and not the binding affinity score. so how did the authors manage to get the score and also what is the unit of these scores?

->Jacalin-RBD complex was allowed to dock with hACE2 and the top model was selected

->what criteria was followed to select the top mode?

->The α -helical, β -strand, (There are many typographical and grammatical errors observed in this paper).

->Even though the authors have mentioned the domains in colour format, still they need to be labelled in the figure to make it more understandable

In order to explore RBD recognizing property of jacalin in drug delivery, we analyzed the ability of jacalin to carry hydroxychloroquine

->Authors need to reframe the sentence. Here the authors have performed blind docking, which means, the grid box was covering the complete protein. Furthermore, the authors have only specified the binding energy and not the inhibition constant which is also very important

->The free energy values suggest stronger binding, which is comparable to those observed for the interaction of carbohydrate and lectin (23)

->Here the authors have mentioned the binding as stronger(Which means they have compared this value with another value). But the other value is not mentioned between the carbohydrate and lectin with their ligand.

These interactions are presented as ligplot in Fig. 4, which suggest that jacalin can serve as a drug carrier

-> how did the authors come to this conclusion.

No therapeutic drugs/vaccine against COVID-19 drives us to propose jacalin.

->This statement is too straight

In order to explore RBD recognizing property of jacalin in drug delivery, we analyzed the ability of

jacalin to carry hydroxychloroquine.

->it is quite confusing, that why the author is using jacalin as a drug carrier, when HCQ can be directly docked with human ACE2 active site

HCQ with the jacalin.

->author should give a valid reason for considering HCQ against jacalin. now this study has three proteins Jacalin-RBD complex with hACE2 to which HCQ is docked. now the changes brought in by HCQ will alter only Jacalin. It is quite unclear, how this will effect the binding between the RBD and ACE2. Authors should have considered only Jacalin with RBD.

Regarding the supplementary material

Table S2: it should be atoms involved and not molecules involved as the authors have only mentioned about the atoms from the respective amino acids

As per Table S1, the jacalin binding site in RBD is much away from the ACE2 binding site. so it may/may not have direct impact on the RBD and ACE2 binding. Then how come there is a change in the Buried Surface Area observed in Table2 and 3. Authors need to explain the reason behind the change in the overall surface area

Decision letter (RSOS-200844.R0)

Dear Dr Veerappan:

Title: Targeting the glycan of receptor binding domain with jacalin as a novel approach to developing treatment against COVID-19

Manuscript ID: RSOS-200844

The editor assigned to your manuscript has now received comments from reviewers. We would like you to revise your paper in accordance with the referee and Subject Editor suggestions which can be found below (not including confidential reports to the Editor). Please note this decision does not guarantee eventual acceptance.

Please submit your revised paper before 15-Jul-2020. Please note that the revision deadline will expire at 00.00am on this date. If we do not hear from you within this time then it will be assumed that the paper has been withdrawn. In exceptional circumstances, extensions may be possible if agreed with the Editorial Office in advance. We do not allow multiple rounds of revision so we urge you to make every effort to fully address all of the comments at this stage. If deemed necessary by the Editors, your manuscript will be sent back to one or more of the original reviewers for assessment. If the original reviewers are not available we may invite new reviewers.

RSC Associate Editor:
Comments to the Author:
(There are no comments.)

RSC Subject Editor:
Comments to the Author:
(There are no comments.)

Reviewers' Comments to Author:
Reviewer: 1

Comments to the Author(s)

In the present manuscript, the authors have tried to verify whether the jacalin can be used as an inhibitor against RBD of Covid-19. Nevertheless, the manuscript lacks in the following aspects:

1. The two docking methods to be compared and whether there are significant changes in the predicted conformations between the top models to be discussed.
2. In the docked complexes, NAGs of hACE2 completely moved away from the protein. Is this the reason for the observed conformational changes rather the binding of jacalin? This might seriously affect the calculated binding affinities. Moreover, the authors need to justify the presence of NAG during docking between hACE2 and RBD-jacalin complex.
3. What are the validation methods followed to choose the right complex apart from docking scores? (both in the case of protein-protein and protein-ligand complexes).
4. Further the authors may dock hACE2 with RBD and compare with the PDB structure to validate the docking protocol.
5. From the final jacalin-RBD-hACE2 complex, it is understandable that jacalin does not interact with the interface between RBD-hACE2 complex. Then, how do the authors justify its efficiency to prevent the interaction of viral protein with host?
6. Are the authors sure that the interactions found would be stable? Molecular dynamics simulation might help the authors to assess the stability of the predicted complexes.

Reviewer: 2

Comments to the Author(s)

This study is quite interesting wherein the authors have tried to dock jacalin with RBD which is in direct contact with hACE2. However, Jacalin was also docked with Hydroxychloroquine. Here onward the study becomes quite laborious to understand

to begin with:

-> The abstract is too short, which does not give any details about the findings

->Regarding the graphical abstract it is mentioned as COVID19 RBD interacting with hACE2.

This is incorrect. Actually COVID19 is the infection and the protein is SARS-Cov-2. So the authors should have mentioned it as SARS-Cov-2 instead

->The statement like "The interaction between a lectin, jacalin and RBD of COVID-19 Coronaviruses (CoV) belongs to a family of enveloped viruses that causes diseases in birds and mammals" is incorrect

->With respect to Materials and methods, it is too short and does not explain the different parameters like active site/interface residues and their reference for protein-protein and protein ligand interaction. Even the grid box for Autodock software is not mentioned in the manuscript. The downloaded 3D structure has no structure validation report.

Regarding Result and discussion:

Authors have added literature review in this section also which need to be removed.

In order to establish the interaction between the glycation site of RDB or hACE2 with jacalin, we used two different protein-protein docking servers, ClusPro2.0 and HADDOCK (High Ambiguity Driven bimolecular DOCKing). ClusPro2.0 performs three steps computational, (i) rigid body docking using the Fast Fourier Transform (FFT) correlation approach; (ii) Root Mean Square Deviation (RMSD) based clustering of the structures generated to find the largest cluster that will represent the likely models of the complex; (iii) refinement of selected structures.

-> this statement is quite confusing.

->Within the Result and discussion session, software details are not required. This should be shifted to materials and methods section

-> Regarding the the score obtained for jacalin-RBD complex and jacalin-hACE2 complex was -82.6 and -78.7, respectively the authors haven't mentioned the +/- value of the HADDOCK score and the significance of the buried surface value

-> The weighted score (balanced model) obtained from ClusPro 2.0 for the jacalin-RBD and jacalin-hACE2 complexes were -1308.8 and -1184.5, respectively.

-> Basically Cluspro only provides the docked poses and not the binding affinity score. so how did the authors manage to get the score and also what is the unit of these scores?

-> Jacalin-RBD complex was allowed to dock with hACE2 and the top model was selected

-> what criteria was followed to select the top mode?

-> The α -helical, β -strand, (There are many typographical and grammatical errors observed in this paper).

-> Even though the authors have mentioned the domains in colour format, still they need to be labelled in the figure to make it more understandable

In order to explore RBD recognizing property of jacalin in drug delivery, we analyzed the ability of

jacalin to carry hydroxychloroquine

-> Authors need to reframe the sentence. Here the authors have performed blind docking, which means, the grid box was covering the complete protein. Furthermore, the authors have only specified the binding energy and not the inhibition constant which is also very important

-> The free energy values suggest stronger binding, which is comparable to those observed for the interaction of carbohydrate and lectin (23)

-> Here the authors have mentioned the binding as stronger (Which means they have compared this value with another value). But the other value is not mentioned between the carbohydrate and lectin with their ligand.

These interactions are presented as ligplot in Fig. 4, which suggest that jacalin can serve as a drug carrier

-> how did the authors come to this conclusion.

No therapeutic drugs/vaccine against COVID-19 drives us to propose jacalin.

-> This statement is too straight

In order to explore RBD recognizing property of jacalin in drug delivery, we analyzed the ability of

jacalin to carry hydroxychloroquine.

-> it is quite confusing, that why the author is using jacalin as a drug carrier, when HCQ can be directly docked with human ACE2 active site

HCQ with the jacalin.

-> author should give a valid reason for considering HCQ against jacalin. now this study has three proteins Jacalin-RBD complex with hACE2 to which HCQ is docked. now the changes brought in by HCQ will alter only Jacalin. It is quite unclear, how this will effect the binding between the RBD and ACE2. Authors should have considered only Jacalin with RBD.

Regarding the supplementary material

Table S2: it should be atoms involved and not molecules involved as the authors have only mentioned about the atoms from the respective amino acids

As per Table S1, the jacalin binding site in RBD is much away from the ACE2 binding site. so it may/may not have direct impact on the RBD and ACE2 binding. Then how come there is a change in the Buried Surface Area observed in Table2 and 3. Authors need to explain the reason behind the change in the overall surface area

Author's Response to Decision Letter for (RSOS-200844.R0)

See Appendix A.

RSOS-200844.R1 (Revision)

Review form: Reviewer 1

Is the manuscript scientifically sound in its present form?

Yes

Are the interpretations and conclusions justified by the results?

Yes

Is the language acceptable?

Yes

Do you have any ethical concerns with this paper?

No

Have you any concerns about statistical analyses in this paper?

No

Recommendation?

Accept as is

Comments to the Author(s)

The authors have addressed all the suggestions in the revised manuscript.

Review form: Reviewer 2

Is the manuscript scientifically sound in its present form?

Yes

Are the interpretations and conclusions justified by the results?

Yes

Is the language acceptable?

No

Do you have any ethical concerns with this paper?

No

Have you any concerns about statistical analyses in this paper?

No

Recommendation?

Accept with minor revision (please list in comments)

Comments to the Author(s)

Authors have incorporated all the changes suggested in the manuscript. But certain typographical errors and grammatical errors are there which needs to be looked into.

There are four structural proteins are identified, which includes spike (S), envelope (E), membrane (M), and nucleocapsid (N) proteins.

->reframe the sentence

e. It infected millions of people and thousands have died (as of -----).

Materials and methods:

ClusPro2.0 performs three steps computational, (i)

->reframe the sentence

The HADDOCK tool is one of the highly utilized and cited docking tools in a single year. Similarly, ClusPro is also a standard tool with over 200 citations.

->reference is missing

pubchem database

->PubChem

box dimension of $22.5 \text{ \AA} \times 22.5 \text{ \AA} \times 22.5 \text{ \AA}$

-> for for x, y and z respectively.

The crystal structure of novel corona spike RBD-ACE2 receptor (PDB ID: 6LZG) with 2.5 \AA

-> 2.5 resolution

To validate the docking procedure, the docking between hACE2 and RBD was performed (separately for both methods) and compared the predicted top model with the PDB structure.

-> compared the predicted top model complex with the PDB structure (as it is docked)

It is clear from HADDOCK and ClusPro 2.0 analysis that the jacalin has slightly higher preference for RBD than hACE2, which may useful in selective targeting the COVID-19.

->which may be useful

Results and discussion

It is noted from Table 3, jacalin affects the binding forces between RBD and hACE2, which clearly reflected in the decreased buried surface area from 2026.1 to 1795.7 \AA^2

->reframe the sentence (binding affinity and not binding forces)

Line 44: All other paramets

->typographical error

The ability of jacalin to interact HCQ was investigated by Autodock Vina server.

-> Ability to interact with HCQ

->

In addition, hydrophobic interaction was noted between HCQ and T72, V81, F104, L106, D125 and Y126.

-> reframe the sentence

2 The spike protein of COVID-19 is a glycoprotein, and jacalin recognize RBD through glycosylation at N343 and has the ability to carry drug.

-> Ability to interact with drug

Decision letter (RSOS-200844.R1)

Dear Dr Veerappan:

Title: Targeting the glycan of receptor binding domain with jacalin as a novel approach to develop treatment against COVID-19

Manuscript ID: RSOS-200844.R1

Thank you for submitting the above manuscript to Royal Society Open Science. On behalf of the Editors and the Royal Society of Chemistry, I am pleased to inform you that your manuscript will be accepted for publication in Royal Society Open Science subject to minor revision in accordance with the referee suggestions. Please find the reviewers' comments at the end of this email.

The reviewers and handling editors have recommended publication, but also suggest some minor revisions to your manuscript. Therefore, I invite you to respond to the comments and revise your manuscript.

Because the schedule for publication is very tight, it is a condition of publication that you submit the revised version of your manuscript before 19-Aug-2020. Please note that the revision deadline will expire at 00.00am on this date. If you do not think you will be able to meet this date please let me know immediately.

1) A text file of the manuscript (tex, txt, rtf, docx or doc), references, tables (including captions) and figure captions. Do not upload a PDF as your "Main Document".

- 2) A separate electronic file of each figure (EPS or print-quality PDF preferred (either format should be produced directly from original creation package), or original software format)
- 3) Included a 100 word media summary of your paper when requested at submission. Please ensure you have entered correct contact details (email, institution and telephone) in your user account
- 4) Included the raw data to support the claims made in your paper. You can either include your data as electronic supplementary material or upload to a repository and include the relevant doi within your manuscript
- 5) All supplementary materials accompanying an accepted article will be treated as in their final form. Note that the Royal Society will neither edit nor typeset supplementary material and it will be hosted as provided. Please ensure that the supplementary material includes the paper details where possible (authors, article title, journal name).

Kind regards,
Dr Laura Smith
Publishing Editor, Journals

RSC Associate Editor:
Comments to the Author:
(There are no comments.)

RSC Subject Editor:
Comments to the Author:
(There are no comments.)

Reviewer comments to Author:
Reviewer: 2
Comments to the Author(s)

Authors have incorporated all the changes suggested in the manuscript. But certain typographical errors and grammatical errors are there which needs to be looked into.

There are four structural proteins are identified, which includes spike (S), envelope (E), membrane (M), and nucleocapsid (N) proteins.

->reframe the sentence

e. It infected millions of people and thousands have died (as of -----).

Materials and methods:

ClusPro2.0 performs three steps computational, (i)

->reframe the sentence

The HADDOCK tool is one of the highly utilized and cited docking tools in a single year. Similarly, ClusPro is also a standard tool with over 200 citations.

->reference is missing

pubchem database

->PubChem

box dimension of $22.5 \text{ \AA} \times 22.5 \text{ \AA} \times 22.5 \text{ \AA}$

-> for for x, y and z respectively.

The crystal structure of novel corona spike RBD-ACE2 receptor (PDB ID: 6LZG) with 2.5 \AA

-> 2.5 \AA resolution

To validate the docking procedure, the docking between hACE2 and RBD was performed (separately for both methods) and compared the predicted top model with the PDB structure.

-> compared the predicted top model complex with the PDB structure (as it is docked)

It is clear from HADDOCK and ClusPro 2.0 analysis

that the jacalin has slightly higher preference for RBD than hACE2, which may useful in selective targeting the

COVID-19.

->which may be useful

Results and discussion

It is noted from Table 3, jacalin affects the binding forces between RBD and hACE2, which clearly reflected in the decreased buried surface area from 2026.1 to 1795.7 \AA^2

->reframe the sentence (binding affinity and not binding forces)

Line 44: All other paramets

->typographical error

The ability of jacalin to interact HCQ was investigated by Autodock Vina server.

-> Ability to interact with HCQ

->

In addition, hydrophobic interaction was noted between HCQ and T72, V81, F104, L106,

D125 and Y126.

-> reframe the sentence

2 The spike protein of COVID-19 is a glycoprotein, and jacalin recognize RBD through glycosylation at N343 and has the ability to carry drug.

-> Ability to interact with drug

Reviewer: 1

Comments to the Author(s)

The authors have addressed all the suggestions in the revised manuscript.

Author's Response to Decision Letter for (RSOS-200844.R1)

See Appendix B.

Decision letter (RSOS-200844.R2)

Dear Dr Veerappan:

Title: Targeting the glycan of receptor binding domain with jacalin as a novel approach to develop a treatment against COVID-19

Manuscript ID: RSOS-200844.R2

It is a pleasure to accept your manuscript in its current form for publication in Royal Society Open Science. The chemistry content of Royal Society Open Science is published in collaboration with the Royal Society of Chemistry.

COVID-19 rapid publication process:

We are taking steps to expedite the publication of research relevant to the pandemic. If you wish, you can opt to have your paper published as soon as it is ready, rather than waiting for it to be published the scheduled Wednesday.

This means your paper will not be included in the weekly media round-up which the Society sends to journalists ahead of publication. However, it will still appear in the COVID-19 Publishing Collection which journalists will be directed to each week (<https://royalsocietypublishing.org/topic/special-collections/novel-coronavirus-outbreak>).

If you wish to have your paper considered for immediate publication, or to discuss further, please notify openscience_proofs@royalsociety.org and press@royalsociety.org when you respond to this email.

RSC Associate Editor
Comments to the Author:
(There are no comments.)

Reviewer(s)' Comments to Author:

Appendix A

Dear Professor Laura Smith,

Thank you for your e-mail message dated 22-06-2020, regarding our manuscript. I am happy to note that it was reviewed favorably and a revision has been suggested.

I am now submitting a revised version of our manuscript wherein we addressed all the points raised by the reviewers.

I hope the present version is suitable for publication in Royal Society Open Science.

I look forward for your positive response.

Sincerely,
V. Anbazhagan

Answer to Reviewers' Comments:

Reviewer: 1

We thank the reviewer for the positive comments and useful suggestions to improve the manuscript.

Comments to the Author(s)

In the present manuscript, the authors have tried to verify whether the jacalin can be used as an inhibitor against RBD of Covid-19. Nevertheless, the manuscript lacks in the following aspects:

1. The two docking methods to be compared and whether there are significant changes in the predicted conformations between the top models to be discussed.

No significant changes were observed in the predicted conformations between the top models obtained from two different docking methods

We have performed two different protein-protein docking methods (ClusPro, and HADDOCK) to prove our objective. Initially, the docking was performed (separately for both methods) between hACE2 and RBD in order to compare the predicted model with the PDB structure. Interestingly, top model (selected based on each method top scoring function) matches with PDB structure and also observed similar interactions between the residues involved. This validates the docking approach.

2. In the docked complexes, NAGs of hACE2 completely moved away from the protein. Is this the reason for the observed conformational changes rather the binding of jacalin? This might seriously affect the calculated binding affinities. Moreover, the authors need to justify the presence of NAG during docking between hACE2 and RBD-jacalin complex.

Thank you very much for pointing out that NAGs of hACE2 was moved away in the top model. The suggestion given by the reviewer was taken sincerely and reworked the docking procedure. In the original version, we considered that the NAGs of hACEs have no role in the protein-protein interaction, because the NAGs of hACE2 were far from the binding regions of our complex, thereby making a very less influence. To support our claim, we did additional experiments using two different models and performed docking between hACE2 and RBD-jacalin complex. (i) Docking jacalin-RBD complex with hACE2 contains NAGs (as reported in original version); (ii) Docking jacalin-RBD complex with hACE2 contains no NAGs; (iii)

Docking jacalin-RBD complex with hACE2 contains NAGs where a distant restraint for NAGs is imposed. This procedure maintains the attachment of NAGs with hAGE2.

The results of the three approaches are given below

	Jacalin-RBD complex-hACE2 (with NAGs)	Jacalin-RBD complex-hACE2 (without NAGs)	Jacalin-RBD complex-hACE2 (with NAGs along with distance restraint)
Cluster rank	1	1	1
Cluster size	354	199	200
HADDOCK score* (a.u)	-131.0 ± 1.5	-134.9 ± 5.2	+4.4 ± 91
RMSD	0.6 ± 0.4	0.7 ± 0.5	1.8 ± 1.6
E _{vdw} (kcal mol ⁻¹)	-59.7 ± 4.2	-66.9 ± 4.6	-61.2 ± 4.2
E _{elec} (kcal mol ⁻¹)	-217.8 ± 38.6	-248.8 ± 20.3	-230.4 ± 38.1
E _{desol} (kcal mol ⁻¹)	1.9 ± 1.55	4.8 ± 0.83	1.9 ± 1.55
E _{AIR} (kcal mol ⁻¹)	-27.9 ± 9.4	-18.7 ± 4.2	1129.4 ± 112.04
Buried Surface Area (Å ²)	1795.7 ± 106.4	1934.0 ± 37.9	1755.7 ± 55.1

It is clear from the table, the main energy components such as E_{vdw}, E_{elec}, E_{desol} showed no difference between the complexes. The only difference seen in E_{AIR} which cannot be quantified for any actual binding energy parameter but attributed to the restraint component added to NAGs. This analysis support that the presence or absence of NAGs in hACE2 does not influence the formation of complex. Thus, we added a sentence to figure 2 legend “NAGs of hACE2 are not involved in the protein-protein interaction”.

3. What are the validation methods followed to choose the right complex apart from docking scores? (both in the case of protein-protein and protein-ligand complexes).

For protein-protein docking: The method was validated by docking hACE2 and RBD and compared the predicted top model with the PDB structure. It was found that the residues involved in the interactions of the top model are similar to the interaction present in the PDB structure. We added this information in the revised version and a fig. S2 is added to the supporting information.

For protein-ligand docking: As suggested by the reviewer, we have validated the docking through one more docking method called ‘DockThor’, a free web server. It was found that jacalin and HCQ binds the same pocket or active site of jacalin found by Autodock tool. The result from the study suggests that jacalin has the ability to carry other important drugs, and this active site in jacalin needs to be noted to enable binding of other important drugs in future.

4. Further the authors may dock hACE2 with RBD and compare with the PDB structure to validate the docking protocol.

As suggested by the reviewer docking of hACE2 with RBD was performed and results are presented as ligplot in Fig. S2. It is noted from the ligplot that the residues involved in the interaction found by docking procedure was as same as PDB crystal structure.

Figure S2: Ligplot diagram showing the interaction between ACE2 and RBD protein structures. (A) From crystal structure complex (PDB Id: 6LZG); (B) From HADDOCK top model structure complex. Similar residues involved in A and B complexes are encircled.

5. From the final jacalin-RBD-hACE2 complex, it is understandable that jacalin does not interact with the interface between RBD-hACE2 complex. Then, how do the authors justify its efficiency to prevent the interaction of viral protein with host?

From the docking result, the change in the secondary structure of the top model was analyzed by DSSP approach. It noted that the binding of jacalin to RBD introduces conformational changes in RBD which in turn proposed to affects the interaction of RBD with hACE2. This was supported by interaction energies and conformational analysis in this study. Moreover, the results show that Jacalin can be used as efficient drug carrier for targeted drug delivery, which opens the possibility of any future drug which could be efficiently targeted using specific lectins.

6. Are the authors sure that the interactions found would be stable? Molecular dynamics simulation might help the authors to assess the stability of the predicted complexes.

Thanks for the suggestion. Current study aimed to show that the lectins can be explored to target RBD. The results obtained from the reported docking procedure were found similar to the reported RBD-hACE2 crystal structure, which was taken as a validation. Moreover, the two different docking programs provided similar results and the consensus in the interactions among all the procedures suggests a stable complex. We agree with the reviewer that the molecular dynamics simulation done in an appropriate time scale will allow us to interpret the stability of the complex; however, further experimental evidences are required to support the MD simulation, which may be done in future. From the obtained results, we try to suggest a novel avenue based on lectins which could be taken up and further validated experimentally to address the ongoing global pandemic.

Reviewer: 2

Comments to the Author(s)

This study is quite interesting wherein the authors have tried to dock jacalin with RBD which is in direct contact with hACE2. However, Jacalin was also docked with Hydroxychloroquine. Here onward the study becomes quite laborious to understand

We thank the reviewer for the positive comments and useful suggestions to improve the manuscript.

to begin with:

-> The abstract is too short, which does not give any details about the findings
 The manuscript was written in communication format, hence brief abstract is given, which illustrate the key point of the study, jacalin has the ability to interact with RBD of SARS-CoV2.

->Regarding the graphical abstract it is mentioned as COVID19 RBD interacting with hACE2. This is incorrect. Actually COVID19 is the infection and the protein is SARS-Cov-2. So the authors should have mentioned it as SARS-Cov-2 instead

	Active residues (Input given in HADDOCK)	
	First molecule	Second molecule
RBD vs Jacalin complex	RBD: 343,342,339,338,368	Jacalin: 1,47,121,122,123,125,78,80
RBD(N343G) vs Jacalin	RBD: 343,342,339,338,368	Jacalin: 1,47,121,122,123,125,78,80
Jacalin-RBD complex vs hACE2	hACE2: 31,19,24,34,41,42,353,83, 30,38,35,82	RBD : 484,475,487,453,500,446,449,498,496,486,505,417,4 93,502,489

Thank you very much for the suggestion. As suggested by the reviewer, graphical abstract is modified.

->The statement like "The interaction between a lectin, jacalin and RBD of COVID-19 Coronaviruses (CoV) belongs to a family of enveloped viruses that causes diseases in birds and mammals" is incorrect

We thank the reviewer for pointing out the error. This was a formatting mistake happen unintentionally. The sentence from the conclusion was accidentally placed. We modified it in the revised version.

->With respect to Materials and methods, it is too short and does not explain the different parameters like active site/interface residues and their reference for protein-protein and protein ligand interaction. Even the grid box for Autodock software is not mentioned in the manuscript. The downloaded 3D structure has no structure validation report.

As suggested by reviewer materials and methods was expanded and the details of active site residues were given. The details of the grid box for Autodock software were included in the revised version.

The 3D structure validation report was obtained from the following PDB.

https://files.rcsb.org/pub/pdb/validation_reports/lz/6lzg/6lzg_full_validation.pdf

https://files.rcsb.org/pub/pdb/validation_reports/m2/1m26/1m26_full_validation.pdf

Regarding Result and discussion:

Authors have added literature review in this section also which need to be removed.

In order to establish the interaction between the glycation site of RDB or hACE2 with jacalin, we used two different protein-protein docking servers, ClusPro2.0 and HADDOCK (High Ambiguity Driven bimolecular DOCKing). ClusPro2.0 performs three steps computational, (i) rigid body docking using the Fast Fourier Transform (FFT) correlation approach; (ii) Root Mean Square

Deviation (RMSD) based clustering of the structures generated to find the largest cluster that will represent the likely models of the complex; (iii) refinement of selected structures.

-> this statement is quite confusing. -> Within the Result and discussion session, software details are not required. This should be shifted to materials and methods section

As suggested by the reviewer, the detail of the software was moved to material and method section.

-> Regarding the the score obtained for jacalin-RBD complex and jacalin-hACE2 complex was -82.6 and -78.7, respectively the authors haven't mentioned the +/- value of the HADDOCK score and the significance of the buried surface value

For discussion we did not used the +/- values, but in Table 3 the +/- values are given.

-> The weighted score (balanced model) obtained from ClusPro 2.0 for the jacalin-RBD and jacalin-hACE2 complexes were -1308.8 and -1184.5, respectively. -> Basically Cluspro only provides the docked poses and not the binding affinity score. so how did the authors manage to get the score and also what is the unit of these scores?

We agree with reviewer that docked poses was obtained from Cluspro. The scores coming from Piper of the model, where the clusters are ranked based on the reweighted score of the top scoring model. The method was used to great success in CAPRI and on various protein docking benchmarks. Selection based on the largest cluster will represent the likely models of the complex. Here, the weighed score is attributed to stable complex, in other words, complex with lowest energy. These are weighed scores obtained from Cluspro, which has no units.

-> Jacalin-RBD complex was allowed to dock with hACE2 and the top model was selected

-> what criteria was followed to select the top mode?

The top model was selected based on top cluster, which usually have lowest energy and RMSD value.

-> The □-helical, □-strand, (There are many typographical and grammatical errors observed in this paper).

This was happen while converting to PDF, which was solved in the revised version. The paper was checked carefully for the other typographical and grammatical errors, and corrected accordingly.

-> Even though the authors have mentioned the domains in colour format, still they need to be labelled in the figure to make it more understandable.

As suggested by the reviewer, the figures are labeled.

In order to explore RBD recognizing property of jacalin in drug delivery, we analyzed the ability of jacalin to carry hydroxychloroquine

-> Authors need to reframe the sentence. Here the authors have performed blind docking, which means, the grid box was covering the complete protein. futhermore, the authors have only specified the binding energy and not the inhibition constant which is also very important.

Thanks for the suggestion. The sentence is reframed and the inhibition constant was added to the revised version.

-> The free energy values suggest stronger binding, which is comparable to

those observed for the interaction of carbohydrate and lectin (23)

->Here the authors have mentioned the binding as stronger(Which means they have compared this value with another value). But the other value is not mentioned between the carbohydrate and lectin with their ligand.

As suggested by the reviewer, we added sentence with additional reference to support the comparison.

These interactions are presented as ligplot in Fig. 4, which suggest that jacalin can serve as a drug carrier-> how did the authors come to this conclusion.

The interaction study suggests that jacalin has binding pocket for the drug. Therefore, it is proposed that it can serve as a drug carrier. For clarity, we remodified the sentence as “These interactions are presented as ligplot in Fig. 4, suggest that jacalin has the binding pocket for hydrophobic drugs, which can be used for the development of jacalin based drug carrier to target the RBD”.

No therapeutic drugs/vaccine against COVID-19 drives us to propose jacalin.

->This statement is too straight

Thanks for the suggestion. The statement is modified as “To support the global research effort of finding solution for COVID-19 drives us to propose jacalin”.

In order to explore RBD recognizing property of jacalin in drug delivery, we analyzed the ability of jacalin to carry hydroxychloroquine. ->it is quite confusing, that why the author is using jacalin as a drug carrier, when HCQ can be directly docked with human ACE2 active site

HCQ was used as model drug to show the drug binding ability of jacalin. This binding pockets can be use to carry other drugs or inhibitors to target the RBD.

HCQ with the jacalin.

->author should give a valid reason for considering HCQ against jacalin. now this study has three proteins Jacalin-RBD complex with hACE2 to which HCQ is docked. now the changes brought in by HCQ will alter only Jacalin. It is quite unclear, how this will effect the binding between the RBD and ACE2. Authors should have considered only Jacalin with RBD.

HCQ was used as model drug to show the drug binding ability of jacalin. These binding pockets can be used to carry other drugs or inhibitors to target the RBD.

Regarding the supplementary material

Table S2: it should be atoms involved and not molecules involved as the authors have only mentioned about the atoms from the respective amino acids

As suggested by the reviewer, it was modified to atoms

As per Table S1, the jacalin binding site in RBD is much away from the ACE2 binding site. so it may/may not have direct impact on the RBD and ACE2 binding. Then how come there is a change in the Buried Surface Area observed in Table2 and 3. Authors need to explain the reason behind the change in the overall surface area

The docking of jacalin-RBD complex with hACE2 affects the secondary structure content of RBD: α -helical (11.28), β -strand (26.15) and coil (62.56), whereas the secondary structure of hACE2 was

unaffected, α -helical (62.75), β -strand (3.52) and coil (33.72). These results clearly indicate that the RBD conformational changes induced by jacalin binding affects the interaction between RBD-hACE2.

Appendix B

Reviewer: 2

Authors have incorporated all the changes suggested in the manuscript. But certain typographical errors and grammatical errors are there which needs to be looked into.

Thank you very much for the suggestion. The suggestions are taken sincerely and corrected in the revised version.

There are four structural proteins are identified, which includes spike (S), envelope (E), membrane (M), and nucleocapsid (N) proteins.

->reframe the sentence

Reframed as “CoV contains four structural proteins, they are spike (S), envelope (E), membrane (M), and nucleocapsid (N) proteins”

e. It infected millions of people and thousands have died (as of -----).

Added “as on August 2020”

Materials and methods:

ClusPro2.0 performs three steps computational, (i)

->reframe the sentence

Reframed as “ClusPro2.0 follows three computational steps, such as”

The HADDOCK tool is one of the highly utilized and cited docking tools in a single year. Similarly, ClusPro is also a standard tool with over 200 citations.

->reference is missing

Reference added

pubchem database

->PubChem

Modified

box dimension of $22.5 \text{ \AA} \times 22.5 \text{ \AA} \times 22.5 \text{ \AA}$

-> for for x, y and z respectively.

Added x, y and z.

The crystal structure of novel corona spike RBD-ACE2 receptor (PDB I D: 6LZG) with 2.5 \AA

-> 2.5 resolution

added

To validate the docking procedure, the docking between hACE2 and RBD was performed (separately for both methods) and compared the predicted top model with the PDB structure.

-> compared the predicted top model complex with the PDB structure (as it is docked)

corrected

It is clear from HADDOCK and ClusPro 2.0 analysis

that the jacalin has slightly higher preference for RBD than hACE2, which may useful in selective targeting the COVID-19.

->which may be useful
corrected

Results and discussion

It is noted from Table 3, jacalin affects the binding forces between RBD and hACE2, which clearly reflected in the decreased buried surface area from 2026.1 to 1795.7 Å²
->reframe the sentence (binding affinity and not binding forces)

Changed as “jacalin affects the binding affinity between RBD and hACE2, as evidenced from the decreased buried surface area from”

Line 44: All other paramets

->typographical error
corrected

The ability of jacalin to interact HCQ was investigated by Autodock Vina server.

-> Ability to interact with HCQ
modified

->In addition, hydrophobic interaction was noted between HCQ and T72, V81, F104, L106, D125 and Y126.

-> reframe the sentence

Reframed as “hydrophobic interaction was identified between”

2 The spike protein of COVID-19 is a glycoprotein, and jacalin recognize RBD through glycosylation at N343 and has the ability to carry drug.

-> Ability to interact with drug
Modified

Reviewer: 1

Comments to the Author(s)

The authors have addressed all the suggestions in the revised manuscript.

We thank the reviewer for the positive comments.